# UBA3 reduction sensitizes cancer cells to NAE inhibitors

Yu-Ling Miao[1,2,*] , Li-Na Zhou[1,2,*], Shan-Shan Song[1], Xu-Bin Bao[1], Xia-Juan Huan[1], Jian Ding[1,2], Jin-Xue He[1,2]

The NEDD8-activating enzyme (NAE), comprising NAE1 and UBA3, is an anticancer target. The relative contribution of each of these subunits toward NAE inhibitor (NAEi) efficacy remains unclear. We demonstrated a profound interdependence of NAE1 and UBA3 expression. UBA3 reduction augmented NAEi sensitivity, whereas its overexpression led to decreased sensitivity. *UBA3* deficiency enhanced RKO xenograft sensitivity to SOMCL-19-133 (NAEi), which was reversed by UBA3 restoration. Cells with naturally low UBA3 expression were highly NAEi-sensitive. The criticality of UBA3 in NAEi sensitivity is not completely unexpected given that the ability of NAEi to directly bind to UBA3 is known. TCGA data showed that rectum adenocarcinoma patients with low *UBA3* mRNA had poorer prognoses, and 27.16% of tumors expressed low *UBA3* mRNA. We propose that low *UBA3* expression may serve as a NAEi sensitivity biomarker, particularly given that MLN4924 (NAEi) phase 3 failures may be due to a lack of patient stratification. Therefore, our key findings, on the criticality of UBA3 in NAEi sensitivity, underpin future clinical evaluations.

## Introduction

Maintenance of protein homeostasis is critical for cell fate ([1]). This process is regulated by the ubiquitin–proteasome system and ubiquitin-like protein (UBL) conjugation pathways ([1]). Neural precursor cell–expressed, developmentally down-regulated protein 8 (NEDD8), an important UBL member, shares a 60% identity with ubiquitin and similarly modifies substrate proteins in a process known as neddylation ([1], [2]). Neddylation regulates the turnover and function of ~20% of cellular proteins. Analogous to ubiquitination, neddylation covalently attaches NEDD8 to substrates via cascade NEDD8-specific E1-E2-E3 enzymatic reactions ([1], [2]). Briefly, NEDD8 is activated by the sole NEDD8-activating enzyme (NAE; E1), then transferred to one of two E2 conjugation enzymes (Ubc12 or UBE2F), and finally conjugated to the target proteins catalyzed by specific E3 ligases ([1], [2]). Substrates of neddylation are broadly classified into two types, which form cullin proteins including cullin-RING ligases (CRLs), a large family of ubiquitin

E3 ligases, and non-cullin effector proteins such as CTD1, p27, p21, pIκBα, and γH2AX ([1], [2]). Neddylation of CRLs targets proteins for proteasomal degradation, whereas modification of non-cullins influences critical cellular processes including DNA replication, DNA damage response, cell cycle arrest, apoptosis, autophagy, and senescence ([1], [2]). Therefore, neddylation plays crucial roles in protein degradation, cell function, and survival.

NAE inhibitors such as MLN4924 and SOMCL-19-133 have demonstrated promising anticancer activity in preclinical/clinical settings ([2], [3], [4], [5], [6], [7]). NAE inhibitors can also cause severe toxicity, particularly hepatotoxicity ([1], [8], [9]), and the NAE inhibitor TAS4464 has been discontinued at the phase 1 clinical stage because of its dose-limiting hepatotoxicity ([8]). The anticancer efficacy and hepatotoxicity of NAE inhibitors are intrinsically linked to the inhibition of their target NAE ([2], [3], [8], [9]). Therefore, how to balance the two sides seems critical for NAE inhibitors' future.

NAE is composed of a regulatory subunit, NAE1, and a catalytic subunit, UBA3 ([10]). Both NAE1 and UBA3 are essential to the survival and growth of cancer cells ([11]). The $Uba3^{O,2,2}$ mutation leads to a mutant lethal phenotype in *Drosophila melanogaster* ([12]). Several rare *NAE1* genetic mutations reduce NAE1 abundance, disrupt the interactions between NAE1 and UBA3, and result in clinical phenotypes ([13]). Within the catalytic center, UBA3 possesses an ATP-binding pocket and a catalytic cysteine residue (Cys237) ([10]). NAE-dependent NEDD8 activation starts from NAE binding NEDD8 and ATP to form a transient NEDD8-AMP intermediate. Subsequently, Cys237 at UBA3 reacts with this intermediate to form a NAE-NEDD8 intermediate. Finally, the latter binds another NEDD8 and ATP to form a NEDD8-NAE-NEDD8-AMP complex that contains an activated NEDD8. NAE inhibitors (e.g., MLN4924) react with the NAE-NEDD8 intermediate to form a relatively stable inhibitor-NEDD8 adduct, thereby blocking NAE catalytic activity and preventing the formation of activated NEDD8. Consequently, neddylation of UBA3 itself, NEDD8-specific conjugating E2s (e.g., Ubc12), CRLs (e.g., Cul1), and effector proteins (e.g., p21, p27, and CDT1) is reduced, significantly impairing NEDD8-dependent signaling transduction and essential intracellular pathways ([1], [10]).

Notably, in the above adducts, NAE1, UBA3, and the NAE inhibitor bind at a molecular ratio of 1:1:1, and the inhibitor directly binds to the ATP-binding pocket of UBA3 ([1], [10]). The former suggests that blocking NAE-dependent NEDD8 activation in cells with lower levels of

---

[1]State Key Laboratory of Drug Research, Cancer Research Center, Shanghai Institute of Materia Medica, Chinese Academy of Sciences, Shanghai, China  [2]University of Chinese Academy of Sciences, Beijing, China

Correspondence: jinxue_he@simm.ac.cn
*Yu-Ling Miao and Li-Na Zhou contributed equally to this work

**Life Science Alliance**

NAE (NAE1 and UBA3) requires fewer inhibitors, indicating that such cells may exhibit increased sensitivity to NAE inhibitors compared to cells with higher NAE levels. The latter implies that the catalytic subunit UBA3 may play a more critical role in determining cellular sensitivity to NAE inhibitors than the regulatory subunit NAE1.

In this study, we confirmed those deductions using in vitro and in vivo preclinical models through RNA interference, gene editing, and exogenous gene expression techniques. The reduction of both NAE1 and UBA3 led to increased cellular sensitivity to NAE inhibitors, whereas their exogenous expression resulted in decreased sensitivity. Though the expression levels of NAE1 and UBA3 were profoundly interdependent, the changes in UBA3 levels had a more pronounced impact on cellular sensitivity to NAE inhibitors. In addition, most of the cancer cells expressing low-level *UBA3* mRNA were highly sensitive to NAE inhibitors (MLN4924 and SOMCL-19-133). In 27.16% solid tumors, the levels of *UBA3* mRNA were significantly lower than those in the corresponding para-cancerous normal tissues. These findings warrant further investigations into low *UBA3* expression as a potential sensitivity biomarker for NAE inhibitors in clinical settings.

## Results

### The expression of NAE1 and UBA3 is profoundly interdependent

The Cancer Cell Line Encyclopedia (CCLE) database revealed a strong positive correlation between the protein abundance of UBA3 and NAE1 in cancer cell lines (Spearman's correlation = 0.86 and $P \ll 0.001$) (Fig 1A). According to this database, both UBA3 and NAE1 are expressed in most of the cancer cell lines (97.05%), and this percentage is very similar to that obtained from the CRISPR/Cas9 analysis in 582 cancer cell lines (11). Notably, only a small fraction of the cell lines exclusively expressed UBA3 (1.61%), NAE1 (1.24%), or neither (0.11%). These data indicate that the co-expression of NAE1 and UBA3 is essential to the survival of most cancer cells. In contrast, there are cells, though very few, that develop alternative NAE-independent survival abilities. Because of the loss of the target, these cells are likely inherently resistant to NAE inhibitors.

To further explore the interdependence of *NAE1* and *UBA3* expression, we separately conducted silencing experiments on *NAE1* and *UBA3* in colon cancer RKO and HCT-116 cells with specific siRNA and shRNA (Fig 1B–E). Transient interference of *NAE1* and *UBA3* using siRNA resulted in an obvious reduction of their respective protein (Fig 1B) and mRNA (Fig 1C). Neddylation of Cul1, a cullin protein substrate (E3 ligase) of NAE, was correspondingly reduced (Figs 1B and S1A). Concurrently, siNAE1 and siUBA3 also caused a notable decrease in the protein levels of their corresponding partner (Fig 1B) while inducing no or only marginal changes at mRNA levels (Fig 1C). The results indicate that the protein levels of NAE1 and UBA3 are mutually influential, although this interaction does not appear to be mediated via a transcriptional mechanism.

The interdependence of NAE1 and UBA3 at protein levels was further corroborated by stable interference using their shRNA. All the specific shRNAs separately targeting *UBA3* and *NAE1* not only

led to the down-regulation of their respective protein levels, but also resulted in the corresponding reduction of their partner protein (Fig 1D and E). Treatments with the proteasome inhibitor MG132 increased the protein levels of NAE1 in the control (shCtrl) cells but did not cause a consistent increase in the cells transfected with either shUBA3 or shNAE1 (i.e., transfection with only one of two respective shRNA sequences resulted in the increase). MG132 did not significantly enhance the protein levels of UBA3 in the shUBA3- or shNAE1-transfected cells (Fig 1D and E). In contrast, the autophagy inhibitor BafA1 did not affect the levels of NAE1 or UBA3 (Fig 1E). These data suggest that the levels of NAE1 and UBA3 are in addition regulated by other possible mechanisms (e.g., complex stability or cotranslational regulation), which remain to be further clarified.

### Reduction of UBA3 sensitizes NAE inhibitors more potently than that of NAE1

In RKO and HCT-116 cells that were stably transfected with either shUBA3 (Fig 2A) or shNAE1 (Fig 2B), reduction of UBA3 (Fig 1D) or NAE1 (Fig 1E) led to increased cellular sensitivity to both NAE inhibitors MLN4924 and SOMCL-19-133. Moreover, the enhancement of NAE inhibitor sensitivity caused by shUBA3 was obviously greater than that caused by shNAE1 (Fig 2A and B). In HCT-116 cells, shUBA3 induced a maximal sensitization of 10-fold, whereas shNAE1 resulted in only fourfold sensitization. Similarly, in RKO cells, shUBA3 led to a maximal 27-fold sensitization, whereas shNAE1 produced only eightfold sensitization (Fig 2A and B). Consistently, the impact of stable shUBA3 transfection on neddylation signaling transduction was more substantial in RKO cells, irrespective of SOMCL-19-133 treatments (Fig 2C). The transfection with shUBA3 not only nearly abolished the neddylation of UBA3, Ubc12, and Cul1 (Figs 2C and S1B), but also resulted in more accumulation of the effector molecules, including CTD1, p27, p21, pIκBα, and γH2AX (Fig 2C). These data indicate that the reduction of either NAE1 or UBA3 protein increases cellular sensitivity to NAE inhibitors, with UBA3 playing a more important role in determining this sensitivity.

### Exogenous expression of UBA3 or NAE1 in RKO parental or UBA3-deficient (UBA3 KO) cells decreases the cellular NAE inhibitor sensitivity

To further investigate the impact of UBA3 and NAE1 on NAE inhibitor sensitivity, we used the CRISPR/Cas9 technology to generate *UBA3*-deficient (UBA3 KO) cells from the RKO cell line. However, the UBA3-deficient cells retained ~30% of UBA3 protein (Fig 3A), possibly because cells that had no UBA3 protein (i.e., complete knockout) could not survive. This result underscores the fundamental essentiality of UBA3 to this cell line. The exogenous expression of *UBA3* in either RKO parental or *UBA3*-deficient cells significantly increased their protein levels of both UBA3 and NAE1 (Fig 3A), and consistently decreased their sensitivity to the NAE inhibitors MLN4924 and SOMCL-19-133 (Fig 3B). In contrast, the exogenous expression of *NAE1* in the same cells significantly enhanced the protein levels of NAE1 (Fig 3A), but only marginally reduced the sensitivity of UBA3-deficient cells to NAE inhibitors

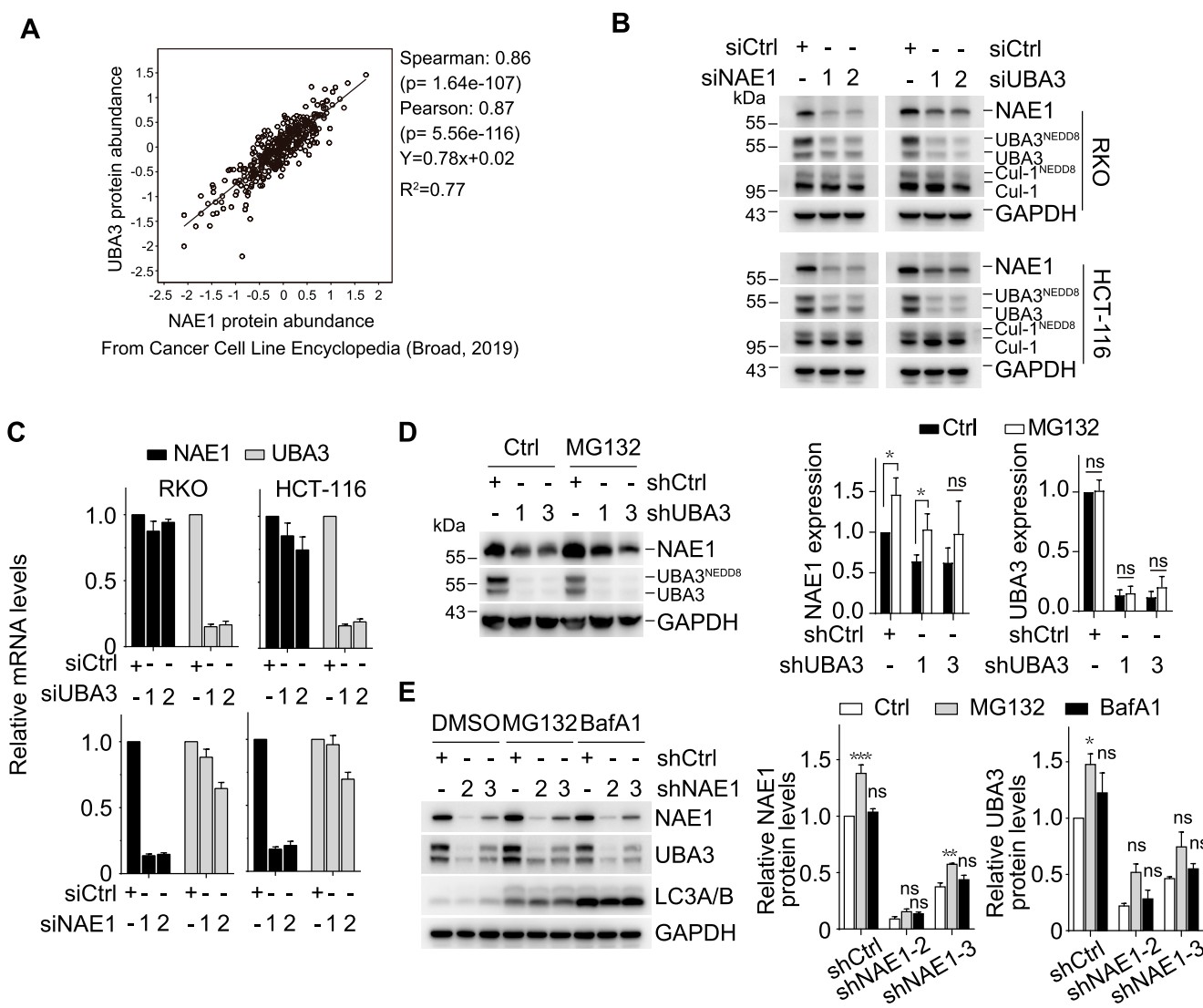

**Figure 1. Expression of *UBA3* and *NAE1* in cells is interdependent.**
**(A)** Correlation between UBA3 and NAE1 protein abundance in cancer cell lines collected in Cancer Cell Line Encyclopedia (CCLE; Broad, 2019). **(B)** Western blotting was done to measure the changes in the levels of NAE1, UBA3, Cul1, and the neddylation products of the latter two in colon cancer RKO and HCT-116 cells exposed to RNA interference with siNAE1 or siUBA3 for 48 h. **(C)** RT–qPCR was conducted to determine the changes in the mRNA levels of *UBA3* (upper) and *NAE1* (lower) in the cells as described in (B). **(D)** Protein levels of NAE1 and UBA3 were measured by Western blotting in shUBA3 or shCtrl RKO cells exposed to MG132 (5 $\mu$M) or DMSO for 12 h (left). The relative pixel intensity from three independent experiments was analyzed with the ImageJ software, and the relative expression of NAE1 and UBA3 was normalized to GAPDH (right). **(E)** Protein levels of NAE1, UBA3, and LC3A/B were examined by Western blotting in shNAE1 and shCtrl RKO cells exposed to MG132 (5 $\mu$M), BafA1 (500 nM), or DMSO for 12 h (left). Semi-quantitative analyses (right) were done as described in (D). In this figure, the numbers 1, 2, and 3 separately represent different siRNA or shRNA sequences targeting the indicated genes. Data were presented as the mean ± SD (n = 3 independent biological replicates). In (D, E), statistical analysis was performed using two-way ANOVA with Šidák's multiple comparison test. *$P < 0.05$; **$P < 0.01$; ***$P < 0.001$; and ns, not significant.
Source data are available for this figure.

(Fig 3B). Notably, the protein levels of both UBA3 and NAE1 in the *UBA3*-exogenously expressed RKO cells were substantially higher than those in the parental RKO cells (i.e., pLEX-Ctrl–transfected cells; Fig 3A), and consistently, the former exhibited higher IC$_{50}$ values of NAE inhibitors (i.e., lower sensitivity) than the latter, indicating that the overexpression of *UBA3* results in reduced NAE inhibitor sensitivity (Fig 3B). However, this phenomenon was not observed in the *NAE1*-overexpressed RKO cells. These data further strengthen the conclusion that UBA3 plays a more

important role than NAE1 in determining cellular NAE inhibitor sensitivity.

### Reduction of UBA3 increases the in vivo sensitivity to SOMCL-19-133, which is reversed by exogenously expressed UBA3

In RKO xenograft models, oral administration of SOMCL-19-133 at doses of 2.5, 5, and 10 mg/kg elicited tumor growth inhibition of 41.58%, 77.26%, and 97.15%, respectively (Figs 4A and S2A),

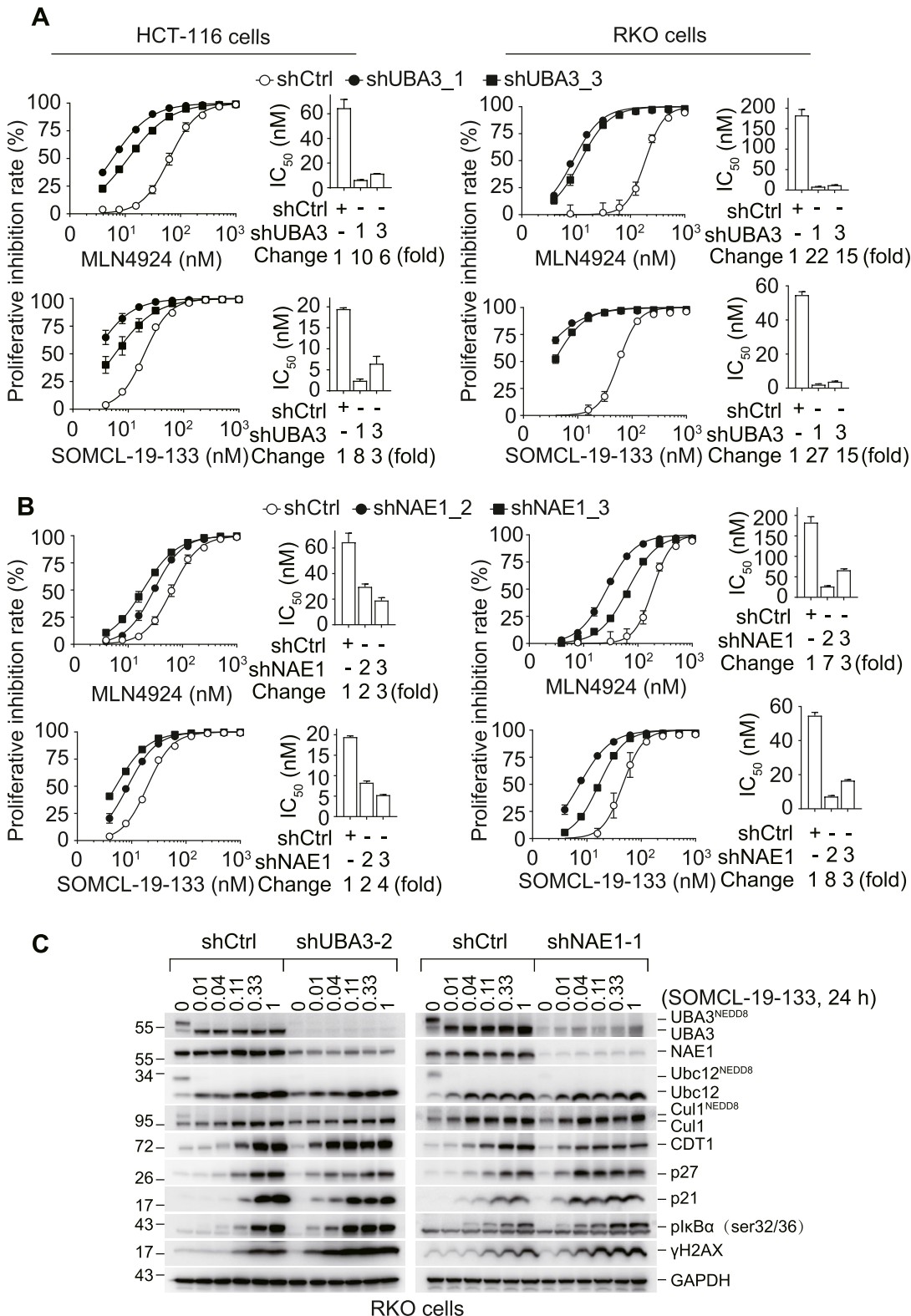

**Figure 2. Reduction of UBA3 sensitizes cancer cells to NAE inhibitors more potently than reduction of NAE1.**
**(A, B)** RKO and HCT-116 cells carrying shUBA3 (A) and shNAE1 (B) were separately treated with MLN4924 or SOMCL-19-133 for 72 h, and the growth inhibition was detected by SRB assays. Concentration–effect curves (left) and $IC_{50}$ values (right) were presented. The change value = $IC_{50/shCtrl}/IC_{50/shUBA3 \text{ or } shNAE1}$. Data were presented as the mean ± SD (n = 3 independent biological replicates). **(C)** RKO cells carrying shCtrl, shUBA3-2, or shNAE1-1 were separately treated with SOMCL-19-133 for 24 h. The

consistent with the previously reported results ([2]). SOMCL-19-133 at 1.25, 2.5, and 5 mg/kg caused tumor growth inhibition of 78.42%, 85.75%, and 99.30% in UBA3 KO xenograft models (Figs 4B and S2B) and 36.21%, 46.91%, and 76.40% in UBA3 KO + UBA3 (exogenous expression of *UBA3* in UBA3 KO) xenograft models (Figs 4C and S2C), respectively. Therefore, SOMCL-19-133 at 1.25 mg/kg in UBA3 KO xenograft models caused tumor growth inhibition almost equal to that elicited by SOMCL-19-133 at 5 mg/kg in either parental RKO or UBA3 KO + UBA3 xenograft models. These findings indicate that *UBA3*-deficient RKO cells result in ~4-fold increase in the in vivo sensitivity to SOMCL-19-133, which is, importantly, almost completely reversed by the exogenous expression of *UBA3*.

In the above experiments, comparable nude mouse models were used. Nevertheless, pharmacokinetic equivalence across models and comparable drug exposure across groups were not demonstrated. Potential host-dependent effects were not assessed. In addition, biochemical toxicity markers (e.g., liver enzymes) were not measured, and changes in body weight of mice alone may not fully capture systemic toxicity. Therefore, caution is warranted when interpreting these results.

### Cancer cells with low expression of *UBA3* are sensitive to the NAE inhibitor SOMCL-19-133

According to the Human Protein Atlas, the average *UBA3* mRNA level across 1,050 cell lines representing 18 cancer types (Fig 5A) was 57.3 nTPM. Of these cell lines, 447 (42.57%), 296 (28.19%), and 173 (16.48%) cell lines exhibited *UBA3* mRNA levels that were 10%, 20%, and 30% lower than the average, respectively. A retrospective analysis of 25 cell lines ([2]), with an average *UBA3* mRNA level of 57.24 nTPM, identified six cell lines (24%) with low *UBA3* mRNA expression (mean, 35 nTPM; range, 31~40 nTPM) as high sensitivity to SOMCL-19-133. The average $IC_{50}$ value for these low-expression cell lines was 3.47-fold lower than that of the remaining 19 cell lines with high *UBA3* mRNA expression (mean, 64 nTPM; range, 42~87 nTPM) (Figs 5B and S3; Table S1). Western blotting analysis demonstrated a strong correlation between *UBA3* protein and mRNA levels across various cancer cell lines (Fig 5C). Further assessment of differential sensitivity to SOMCL-19-133 was conducted on eight cell lines with low *UBA3* mRNA expression (mean: 32 nTPM; range: 13.9~39.9 nTPM) and eight cell lines with high *UBA3* mRNA expression (mean, 67 nTPM; range, 54.7~81.9 nTPM). Despite a mere 2.09-fold difference in *UBA3* mRNA expression, the former exhibited 17.00-fold higher sensitivity to SOMCL-19-133 than the latter (Table S2). Furthermore, human pancreatic cancer SUIT-2 cells had low expression of *UBA3* mRNA and protein, and were sensitive to SOMCL-19-133 (Fig 5C; Table S1). SUIT-2 xenografts in nude mice were also significantly sensitive to SOMCL-19-133 with ~80% inhibition of tumor growth at 10 mg/kg (Fig 5D). These data further indicate that cancer cells with native low expression of *UBA3* are sensitive to the NAE inhibitor SOMCL-19-133, both in vitro

and in vivo. Notably, however, there were no proportional relationships between the expression levels of *UBA3* mRNA and the SOMCL-19-133 sensitivity in those cancer cell lines (Fig S3; Tables S1 and S2), suggesting the involvement of other contributing factors.

## Discussion

NAE1 and UBA3 are the regulatory and the catalytic subunits of NAE, respectively ([10]). Our data indicate that their expression in cells is profoundly interdependent, although the precise mechanisms underlying this interdependence require further elucidation. Moreover, a reduction in the expression of either *NAE1* or *UBA3* has been shown to enhance cellular sensitivity to NAE inhibitors. Notably, UBA3 is proved to play a more prominent role in determining this sensitivity, as evidenced by the significantly greater sensitization observed after UBA3 reduction. In parental RKO cells, the exogenous expression of *UBA3* causes substantial resistance to NAE inhibitors (5.7-fold for SOMCL-19-133 and 7.2-fold for MLN4924), whereas the exogenous expression of *NAE1* does not confer such resistance. Consistent with these findings, the exogenous expression of *UBA3* in the UBA3 KO cells completely reverses the increased sensitivity to NAE inhibitors. These in vitro results are corroborated by in vivo experiments, where *UBA3* deficiency leads to ~4-fold increases in the sensitivity to SOMCL-19-133 in RKO xenograft models, which is almost entirely reversed by *UBA3* reconstitution. Particularly, noteworthy is that the resulting (surviving) cells derived from RKO cells via *UBA3* single-gene knockout (KO) using the CRISPR/Cas9 technology retained ~30% of the UBA3 protein. This result indicates that UBA3 is fundamentally essential to this cell line as observed in other cancer cell lines using the same technology ([11]).

The high NAE inhibitor sensitivity resulting from the reduction of *UBA3* expression has been corroborated in cancer cell lines and xenograft models characterized by inherently low expression of *UBA3*. The retrospective analysis of our previously published data ([2]) reveals that all cell lines exhibiting low *UBA3* mRNA expression are highly sensitive to SOMCL-19-133. Furthermore, an additional eight cell lines with similarly low *UBA3* mRNA levels exhibit comparable sensitivity. Moreover, *UBA3*-lowly expressed SUIT-2 xenografts also display significant sensitivity to SOMCL-19-133.

SOMCL-19-133 exhibits similar NAE inhibition in cell-free systems but demonstrates greater potency in cellular and animal xenograft models than MLN4924 ([2]). However, both the NAE inhibitors possess a highly similar core structure of AMP, one of the NAE substrates (Fig 6A) ([1], [2], [10]). Therefore, these inhibitors can substitute for AMP to form stable inhibitor-NEDD8 adducts, and thereby block the NAE catalytic process (Fig 6B). To form sufficient inhibitor-NEDD8 adducts to block this process, lower levels of UBA3 require fewer molecules of NAE inhibitors (Fig 6C), as the ATP-binding pocket and the catalytic active center (Cys237) are located at UBA3 ([1], [10]). Considering this molecular mechanism, it is

---

levels of the indicated proteins were detected by Western blotting. Data were from three independent experiments, and, where applicable, were expressed as the mean ± SD (n = 3 independent biological replicates).
Source data are available for this figure.

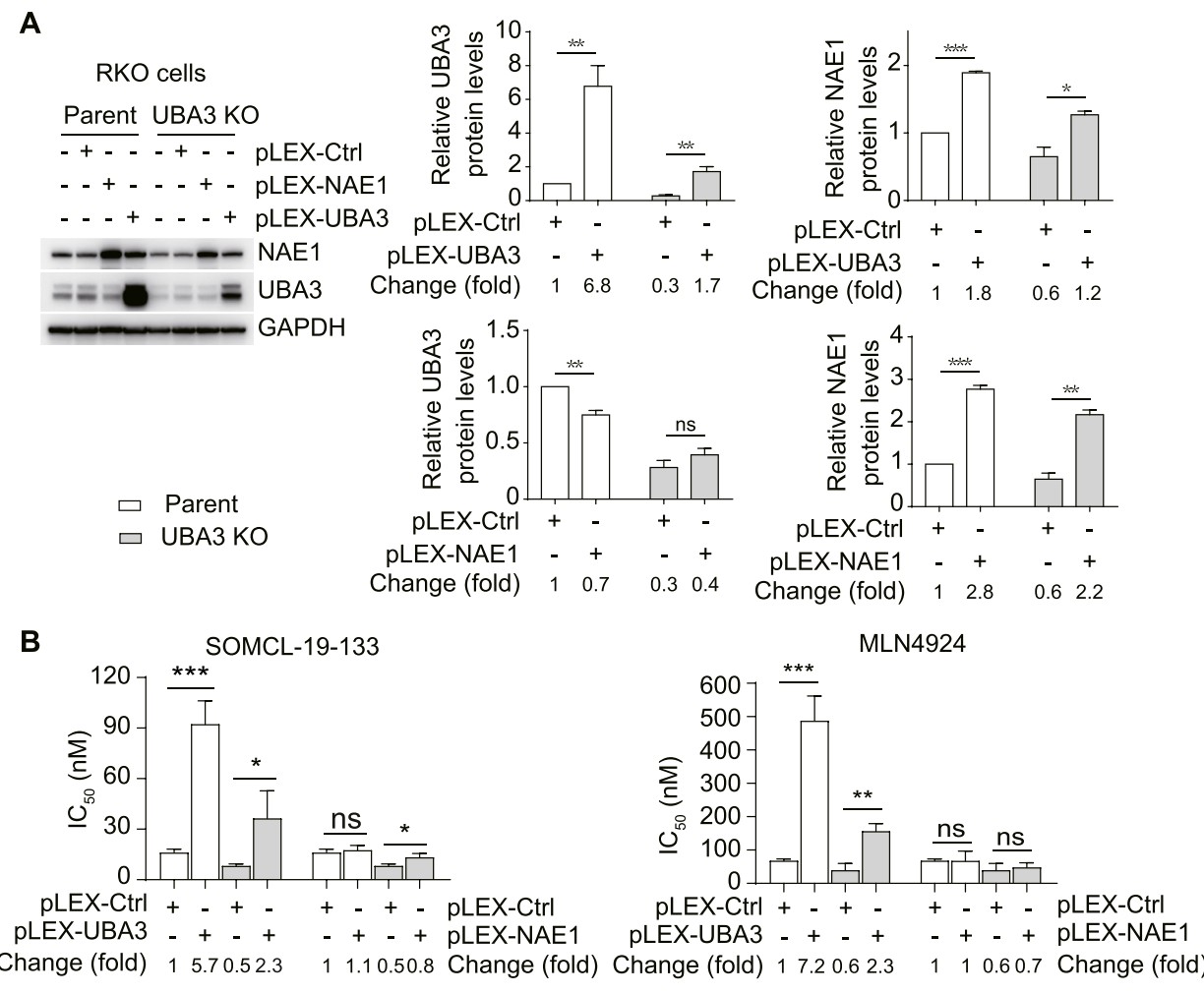

**Figure 3. Exogenous expression of UBA3 or NAE1 in RKO parental or *UBA3*-deficient (UBA3 KO) cells decreases the cellular NAE inhibitor sensitivity.**
**(A)** Levels of the indicated proteins were detected by Western blotting in RKO parental or *UBA3*-deficient (UBA3 KO) cells in which UBA3 or NAE1 was exogenously expressed (left). The semi-quantitative analyses (right four subpanels) were done as described in Fig 1D. **(B)** Cells mentioned above were treated with SOMCL-19-133 or MLN4924 for 72 h and then subjected to SRB assays. Their $IC_{50}$ values were presented separately. The change value = Value$_{/shUBA3\ or\ shNAE1}$/Corresponding value$_{/shCtrl}$. Data are from three independent experiments, and, where applicable, were expressed as the mean ± SD (n = 3 independent biological replicates). Statistical analysis was performed using an unpaired, two-tailed *t* test. *$P < 0.05$; **$P < 0.01$; ***$P < 0.001$; and ns, not significant.
Source data are available for this figure.

comprehensible that cancer cells with either artificially or naturally low expression of *UBA3* are sensitive to these AMP-mimicking NAE inhibitors.

The increased sensitivity of cancer cells with lower levels of UBA3 to NAE inhibitors is also reflected in a more prominent reduction in the neddylation of cullin and some non-cullin substrate proteins. Reduction of UBA3 in RKO cells with shUBA3 enhances the inhibitory effect of the NAE inhibitor SOMCL-19-133 on the neddylation of Ubc12, Cul1 (cullins; components of the neddylation pathway), and led to increased accumulation of CDT1, p27, p21, pIκBα, and γH2AX (non-cullins; effector proteins). These proteins have been extensively demonstrated to play critical roles in the anticancer activity of NAE inhibitors (1, 2, 3). Reduced neddylation of Ubc12 and Cul1 inhibits the neddylation pathway, whereas the stability of those effector proteins increases, resulting in their greater accumulation. Consequently, reduction of UBA3 promotes

NAE inhibitor-induced DNA re-replication, induces DNA damage, and triggers G2 cell cycle arrest and apoptosis. Therefore, the increased sensitivity observed at low UBA3 levels not only arises from altered inhibitor-NAE adduct formation, but also reflects broader changes in the regulation of NAE substrates and downstream signaling pathways.

These findings have important implications for clinical decision-making in cancer therapy. In fact, patients with rectum adenocarcinoma of low *UBA3* mRNA expression have significantly worse prognoses with zero 5-yr survival (Fig 7A). Although the expression of *UBA3* mRNA does not display pronounced tissue specificity (Fig 7B), approximately half of the tumor samples analyzed (49.21%; totaling 6,922 samples across 12 cancer types) exhibit *UBA3* mRNA levels that are lower than those in the corresponding paracancerous normal tissues (Fig 7C). Relative to normal samples, 27.16% of tumor samples (1,880 samples) present *UBA3* mRNA levels

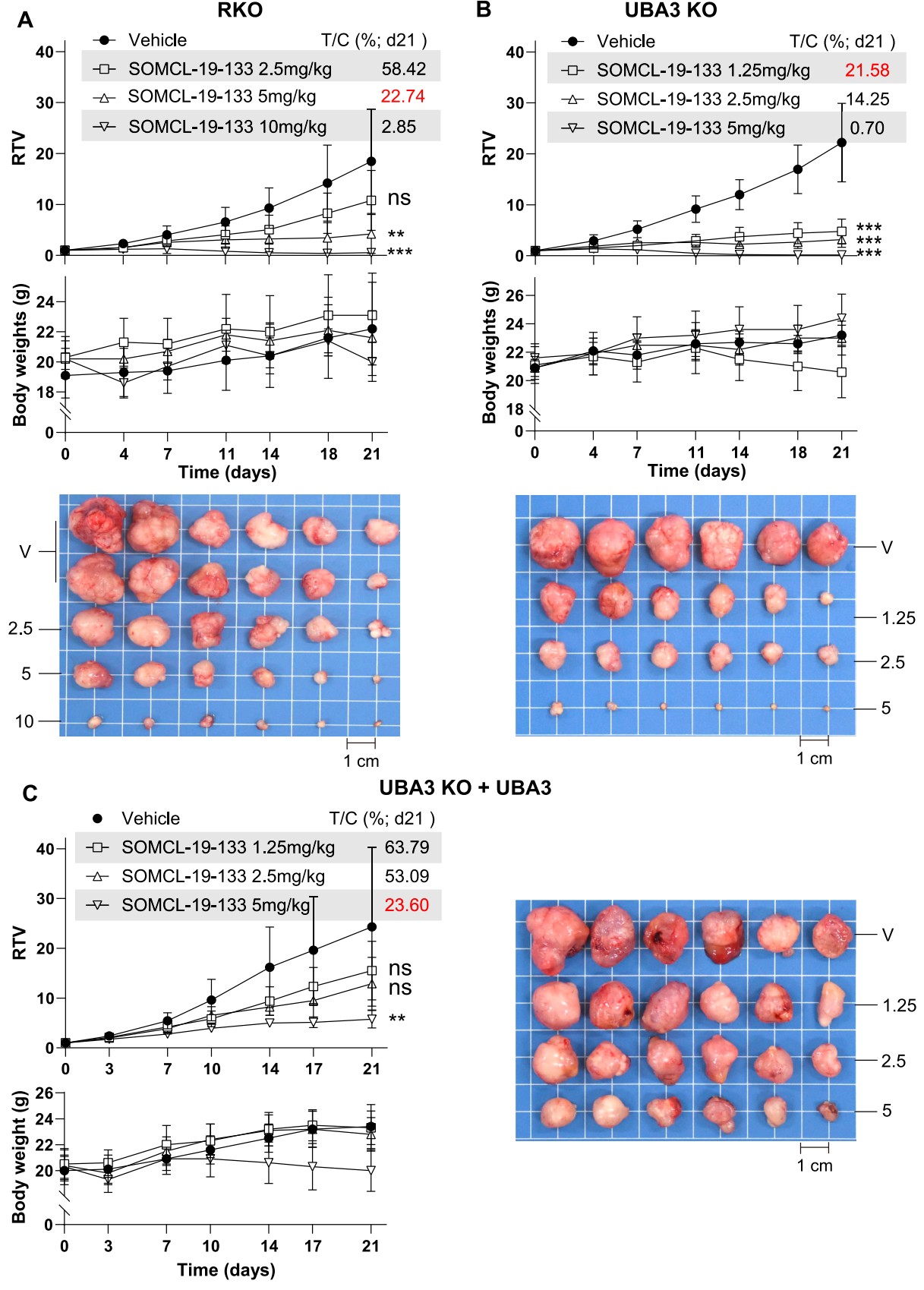

with z-scores < −1. This proportion closely aligns with the 28.19% of cancer cell lines exhibiting *UBA3* mRNA levels ≤40 nTPM in most cases, which are typically 20% lower than the average levels (Fig 5A). Our data reveal that cancer cell lines with *UBA3* mRNA levels ≤40 nTPM are generally sensitive to the NAE inhibitor SOMCL-19-133 (Fig 5B and D; Tables S1 and S2). Notably, certain cancer types demonstrate significantly higher proportions of tumor samples with low *UBA3* mRNA levels (i.e., z-scores < −1, relative to their corresponding paracancerous normal tissues), including head and neck squamous cell carcinoma (64.1%) and renal cancer (43.6%) (Fig 7C). Hepatotoxicity is one of the primary toxicities of NAE inhibitors and is directly related to NAE inhibition (1, 8, 9). Normal hepatic tissues do not express high levels of *UBA3* mRNA (Fig 7B). Nevertheless, there are still 18.2% of hepatocellular carcinoma samples expressing low levels of *UBA3* mRNA (i.e., z-scores < −1) (Fig 7C). To ensure enough safety and satisfy therapeutic efficacy, these data can aid in identifying appropriate cancer patients for treatment with NAE inhibitors.

The NAE inhibitor MLN4924 has been extensively evaluated in both monotherapy and combination therapy settings through phase 1~3 clinical trials (1, 4, 5, 6, 7, 8, 14, 15, 16, 17, 18, 19). Data from phase 1 trials of MLN4924 plus azacitidine showed no obvious increase in toxicities including hepatotoxicity (dose-limiting toxicity) but an apparent enhancement of overall response rate from 27.8% to 50% (4). In phase 2 trials, its combinations with azacitidine (5) or docetaxel (15) and its triplet combination with azacitidine and venetoclax (19) demonstrated encouraging activity in different population of tumor patients. Unfortunately, however, the phase 3 trial of its combination with azacitidine failed to meet the primary therapeutic endpoint (6). Detailed analyses of the data from this phase 3 trial show that patients who underwent prolonged treatment duration (>3 cycles) exhibit a significant difference in overall survival (6). It is posited that excluding patients unable to receive more than three cycles (22.36%; 36/161) (6) at the enrollment stage might have prevented the failure of this phase 3 trial.

The failure of this phase 3 clinical trial highlights the critical importance of appropriate patient selection. Using low *UBA3* expression in cancer as a sensitivity biomarker for NAE inhibitors offers several potential advantages: (1) a substantial proportion of cancer patients accounting for nearly one third of all individuals with solid tumors and over 50% in certain cancer types exhibit low *UBA3* expression; (2) in patients with cancers characterized by low *UBA3* expression, low-dose NAE inhibitors are likely to achieve the desired therapeutic efficacy while significantly reducing toxicity; and (3) the detection of *UBA3* mRNA and protein is both feasible and cost-effective for both hospitals and patients. Our findings provide a critical foundation for further clinical investigations.

However, it is worth noting that mRNA-based stratification may not always reflect functional protein abundance in the clinic, although we have shown a strong correlation relationship between mRNA and protein levels of *UBA3* in cancer cell lines. Tumor

heterogeneity and *UBA3* expression in normal tissues also represent variables that could potentially affect its utility as a biomarker of NAE inhibitor sensitivity. In addition, there is no strict proportional relationship between the expression levels of *UBA3* mRNA and NAE inhibitor sensitivity. Indeed, some cancer cell lines with high *UBA3* mRNA levels remain sensitive to NAE inhibitors (Table S1), indicating that multiple factors contribute to the determination of NAE inhibitor sensitivity. For example, factors such as the vesicular overexpressed in cancer prosurvival protein 1 (VOPP1) (20), mutations in UBA3 (21), and phosphatase and tensin homolog (PTEN) (22) have been reported to influence the anticancer effects of NAE inhibitors. Therefore, further exploration is needed to determine the feasibility of UBA3 as a biomarker and/or to identify different combinations of sensitivity biomarkers for NAE inhibitors.

# Materials and Methods

## Drugs and antibodies

SOMCL-19-133 was prepared as shown in Fig S4 (2). Compound 1 was dissolved in acetonitrile, and then, triethylamine (2 eq) and BOC anhydride (1.05 eq) were added to the solution. After stirring at RT for 3 h, the raw materials were completely reacted. Then, the solvent was dried by rotary evaporation, and Compound 2 was obtained by silica gel column chromatography (DCM: MeOH = 30: 1). Compound 2 was added to a pressure tube filled with dichloromethane. Under the protection of nitrogen, silver oxide (5 eq) and methyl iodide (6 eq) were added to the solution. After heating and reacting at 80°C for 30 min, the raw materials were completely reacted, the mixture was suction-filtered with diatomaceous earth, the filtrate was dried by rotary evaporation, and then Compound 3 was obtained by silica gel column chromatography (PE:EA = 20:1). Compound 3 was dissolved in dichloromethane, and 4 M HCl in dioxane was added to the reaction solution. After 30 min, the solvent was dried by rotary evaporation to obtain Compound 4. Under the protection of nitrogen, Compound H was dissolved in acetonitrile, and Compound 4 (1.3 eq) and DIPEA (3 eq) were added to the solution. The mixture was stirred at RT for 8 h, and the raw materials were completely reacted. Then, the solvent was dried by rotary evaporation, and Compound 5 was obtained by silica gel column chromatography (DCM: MeOH = 30:1). Compound 5 was dissolved in acetonitrile, and then, pyridine (3 eq) and sulfamyl chloride (1.2 eq) were added to the solution at 0°C. After stirring for 2 h, the raw materials were completely reacted. Then, the solvent was dried by rotary evaporation, and SOMCL-19-133 was obtained by silica gel column chromatography. High-performance liquid chromatography (HPLC) analysis confirmed the purity to be ≥98%. [1]H NMR

**Figure 4. *UBA3* deficiency in RKO cells increases, whereas the exogenous expression of *UBA3* in UBA3 KO cells reverses the in vivo sensitivity to SOMCL-19-133. (A, B, C)** Xenografts of RKO parental cells (A), *UBA3*-deficient (UBA3 KO) cells (B), and *UBA3* reconstitution (UBA3 KO + UBA3) cells (C) were treated with SOMCL-19-133 (oral administration) for 21 d. Relative growth rate (RTV), body weight of mice, and xenograft images were presented. Data were, where applicable, expressed as the mean ± SD (n = 12 animals for the RKO vehicle group, and n = 6 animals for all other groups). Scale bars: 1 cm. Statistical analysis was performed using one-way ANOVA. *$P <$ 0.05; **$P <$ 0.01; ***$P <$ 0.001; and ns, not significant.
Source data are available for this figure.

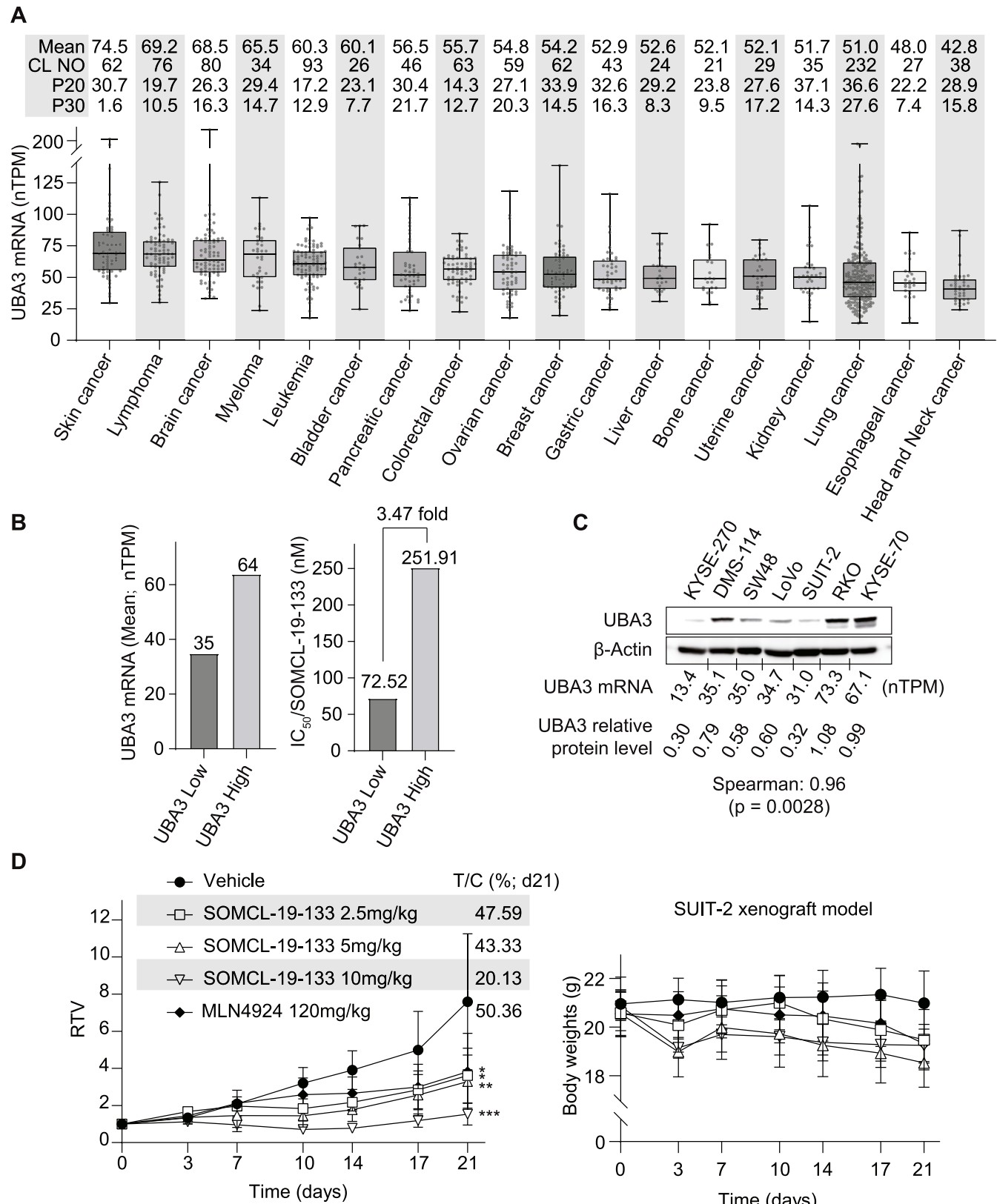

**Figure 5. UBA3 mRNA expression in cancer cells and its association with SOMCL-19-133 sensitivity.**
**(A)** mRNA levels of *UBA3* in cancer cell lines collected in the Human Protein Atlas (https://www.proteinatlas.org/). CL NO, number of cell lines; P20 and P30, percentages of the number of cell lines with *UBA3* mRNA ≤ −20% and −30% mean, respectively. **(B)** Retrospective analysis on the relationship between $IC_{50}$ values of SOMCL-19-133 in 28 cancer cell lines reported in reference 2 and corresponding cellular levels of *UBA3* mRNA from the Human Protein Atlas. The average $IC_{50}$ values of

(500 MHz, methanol-d4) $\delta$ 8.46 (s, 0.76H), 8.31 (s, 0.21H), 7.33–7.16 (m, 4H), 6.62 (d, J = 5.3 Hz, 0.23H), 6.05 (d, J = 5.4 Hz, 0.73H), 5.69–5.60 (m, 1H), 4.62–4.57 (m,1H), 4.43 (dd, J = 9.8, 7.4 Hz, 1H), 4.40–4.34 (m, 1H), 4.26 (dd, J = 9.8, 7.3 Hz, 1H), 3.43–3.36 (m, 1H), 3.27–3.11 (m, 2H), 2.98–2.89 (m, 1H), 2.68–2.60 (m, 1H), 2.51–2.44 (m, 1H), 2.38–2.31 (m, 2H). $^{13}C$ NMR (126 MHz, MeOD) $\delta$ 157.70, 157.12, 156.17, 149.43, 142.15, 141.59, 129.29, 127.95, 126.19, 125.45, 83.52, 73.09, 70.53, 58.13, 57.88, 57.76, 44.80, 43.01, 36.77, 34.38. ESI ([M - H]+) m/z 474.5. HRMS (ESI) calcd for $C_{20}H_{24}N_7O_5S$, 474.1565; found, 474.1578.

MLN4924, MG132, and bafilomycin A1 (BafA1) were purchased from MedChemExpress (Monmouth Junction, NJ). All drugs were dissolved in dimethyl sulfoxide (DMSO; Sigma-Aldrich), aliquoted, and stored at −20°C.

Antibodies against NAE1/APPBP1 (#14321S), Ubc12 (#5641S), Cdt1 (#8064S), p27 (#3688S), pIκBα (ser32/36; #9246S), and LC3A/B (#4108s) were purchased from Cell Signaling Technology. Antibodies against UBA3 (sc-377272), Cul1 (sc-17775), p21 (sc-271610), and γH2AX (sc-101696) were obtained from Santa Cruz Biotechnology. The antibody against GAPDH (AG019) was from Beyotime Biotechnology. The secondary antibodies were from Jackson ImmunoResearch Laboratories.

### Cell lines

Human cancer cell lines NCI-H226, LoVo, SW620, HCT-15, LNCaP clone FGC, VCaP, MDA-MB-231, NCI-N87, HGC-27, and AGS were purchased from the Shanghai Institutes for Biological Sciences at the Chinese Academy of Sciences. DMS-114 was from EK-Bioscience. SW480 was from West China Medical Center, Sichuan University (Chengdu, Sichuan, China). KYSE-270 was from YOBIBIO Biotechnology Co., Ltd. KYSE-70 was from Hunan Fenghui Biotechnology Co., Ltd. CAL27 was from MeilunBio. SUIT-2 was from the Japanese Collection of Research Bioresources Cell Bank (JCRB). All other cell lines used in this study were purchased from the American Type Culture Collection (ATCC). Cells were cultured according to the suppliers' instructions, authenticated via short tandem repeat analysis by Genesky, and tested for *Mycoplasma* contamination.

### RNA interference

Small interfering RNAs (siRNA) targeting *NAE1* and *UBA3* were purchased from RiboBio. Cells were transfected with siRNA oligonucleotides using Lipofectamine RNAiMAX Reagent (Invitrogen) diluted in UltraFectin OptiMEM (BasalMedia) following the manufacturer's instructions. Total RNA extraction was performed 48 h post-transfection, and knockdown of targeted genes was verified via reverse transcription–quantitative polymerase chain reaction (RT–qPCR) and/or Western blotting. The sequences of the sense and antisense strands of the used siRNAs are listed in Table S3. Antibodies

used in Western blotting for verification of reduced protein were those against UBA3 (sc-377272; Santa Cruz Biotechnology) and NAE1/APPBP1 (#14321S; Cell Signaling Technology), respectively.

For the stable knockdown of *NAE1* and *UBA3*, pLKO.1 vectors encoding short hairpin RNAs (shRNA) targeting the indicated genes were transfected into RKO and HCT-116 cells through lentiviral infection. After 72-h lentiviral infection, the clones were selected using 1 μg/ml puromycin for an additional 72 h, and the gene knockdown levels were verified via Western blotting. The sequences of the used shRNA are shown in Table S3. Antibodies used in Western blotting for verification of reduced protein were those against UBA3 (sc-377272) and NAE1/APPBP1 (#14321S), respectively.

### *UBA3* single-gene knockout (KO) via CRISPR/Cas9

*UBA3* single-gene knockout via CRISPR/Cas9 was conducted as previously described (20, 23, 24). pLenti-CRISPRv2 vectors (OBiO Technology) were inserted with different *UBA3*-specific sgRNAs (Table S3). The resulting constructs were cotransfected into 293T cells with psPAX2 and pMD2G packaging plasmids to produce lentiviruses encoding the corresponding sgRNAs. These lentiviruses were separately used to infect colon cancer RKO cells. The infected cells were then subjected to puromycin selection and monoclonal isolation. Gene knockout efficiency was verified by Western blotting. The surviving cells, which retained ~30% of UBA3 protein, were termed *UBA3*-deficient cells or UBA3 KO cells for convenience. The antibody used in Western blotting for verification of reduced protein was that against UBA3 (sc-377272).

### Exogenous expression of *UBA3* and *NAE1*

The exogenous expression of *UBA3* or *NAE1* in RKO and UBA3 KO cells was separately done as previously described (20, 25). In brief, *UBA3* or *NAE1* cDNA was cloned into a pLEX-MCS empty vector, amplified, and then used to produce lentiviruses by OBiO Technology. For the exogenous expression of *UBA3* or *NAE1*, lentiviruses carrying the cDNA sequences of the indicated target genes were seeded into the plates containing the corresponding cancer cells. After infection for 16 h, the cell medium was replaced with fresh medium. After 72 h of infection, the infected cells were then screened with 1 μg/ml puromycin (Sigma-Aldrich) for another 3 d.

### Cell viability assays

Cells were plated at a density of 1,000–8,000 cells per well in 96-well plates before treatment with the indicated drugs at gradient concentrations for 3 d. To measure cell viability, sulforhodamine B (SRB; Sigma-Aldrich) and Cell Counting Kit-8 (CCK-8) assays were performed as follows (26, 27).

---

SOMCL-19-133 in *UBA3* mRNA low and high groups were separately shown. **(C)** Consistency between *UBA3* mRNA (from the Human Protein Atlas) and protein (Western blot) levels. Relative UBA3 protein pixel intensity was quantified using ImageJ and normalized to β-Actin (mean SD, n = 3 independent biological replicates). A correlation analysis between *UBA3* mRNA and protein levels across cell lines was performed. **(D)** Xenografts of the human pancreatic cancer SUIT-2 cell line with low expression of *UBA3* mRNA and protein were significantly sensitive to SOMCL-19-133 (oral administration). Relative growth rate (RTV) (left) and body weight of mice (right) were presented. Data were, where applicable, expressed as the mean ± SD (n = 12 animals for the vehicle group, and n = 6 animals for the treatment groups). Statistical analysis in (D) was performed using one-way ANOVA. *$P < 0.05$; **$P < 0.01$; and ***$P < 0.001$.
Source data are available for this figure.

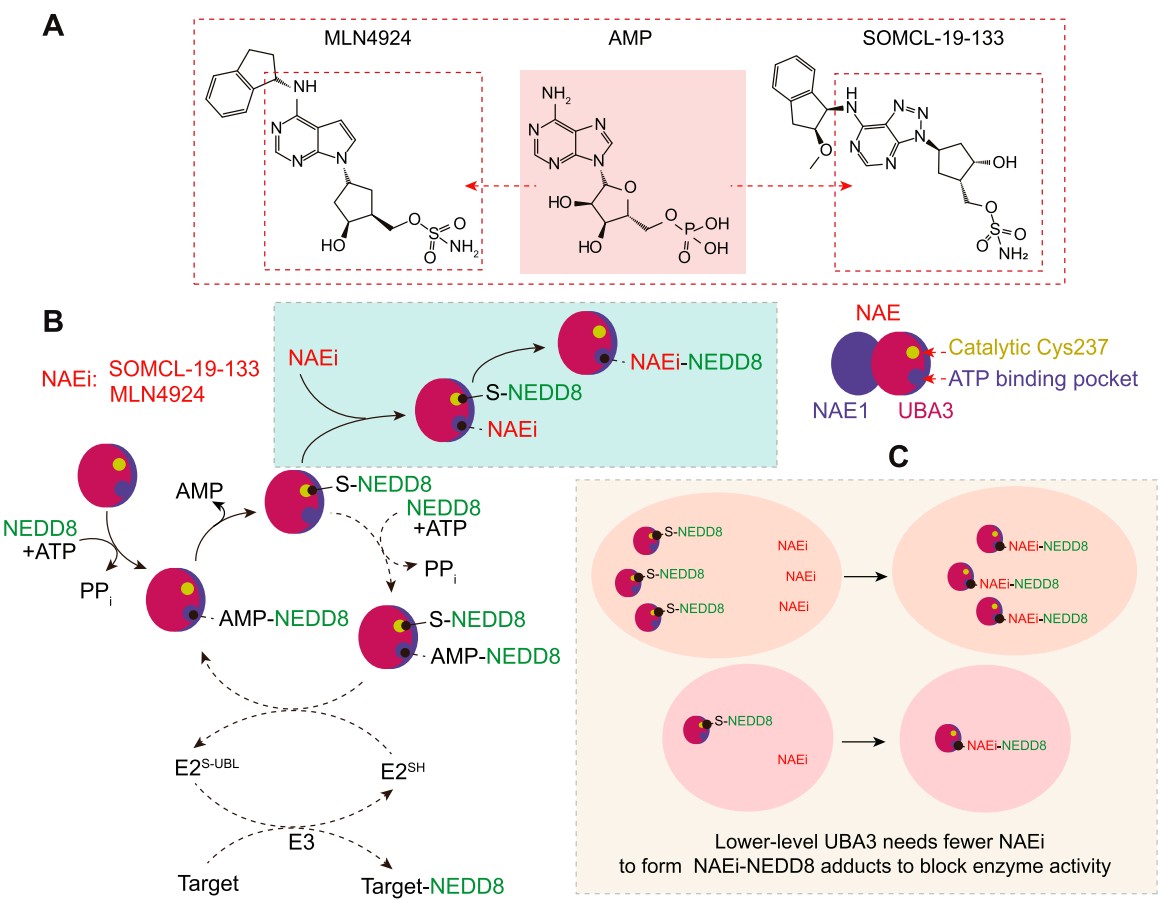

**Figure 6. Simplified mechanism for the possible impact of cellular UBA3 levels on NAE inhibitor sensitivity.**
**(A)** Structural similarity between AMP and NAE inhibitors (NAEi) MLN4924 (3) and SOMCL-19-133 (2). **(B)** Schematic of two subunits of NAE and the ATP-binding pocket and the catalytic cysteine (Cys237) at UBA3 and a simplified mechanism by which NAE inhibitors inhibit NAE, adapted from references 10, 14. **(C)** Schematic of how lower-level UBA3 needs fewer NAEi to form NEDD8-NAEi adducts to block NAE enzyme activity.

To perform SRB assays, cells were fixed with 100 µl/well of trichloroacetic acid (TCA; Sinopharm Chemical Reagent Co., Ltd.) overnight at 4°C. The next day, the TCA was discarded, and the 96-well plate was washed five times with double-distilled water to remove residual TCA. After the plate was completely dried, 100 µl/well of SRB solution was added and incubated for 30 min. The supernatant was then discarded, and the unbound dye was washed away with 1% acetic acid solution. The dried stain was then solubilized with 150 µl/well of 10 mM Tris (Sinopharm Chemical Reagent Co. Ltd.) solution. After SRB was fully dissolved, the plate was placed on an MD SpectraMax 190 microplate reader. The Tris solution containing dissolved SRB in each well was mixed by shaking, and the optical density (OD) was measured at a wavelength of 560 nm.

To perform CCK-8 assays, 10 µl/well of CCK-8 chromogenic reagent was added, and the cells were further incubated in the culture incubator until the OD of the control group cells at a wavelength of 450 nm reached the range of 1.1–1.4. The OD values were recorded using an MD SpectraMax 190 microplate reader.

The inhibition rate (%; IR) was calculated as follows: IR= $(OD_{control} - OD_{treat})/OD_{control} \times 100$. The $IC_{50}$ values were calculated from three independent experiments with the logit method.

## Western blotting

The cellular levels of the indicated proteins were measured using the standard Western blotting as described previously (20, 23). Briefly, cells treated with the indicated drugs were lysed using SDS lysis buffer (50 mM Tris–HCl, pH 6.8, 2% SDS, 0.05% bromophenol blue, 10% glycerin, and 100 mM DTT) and subjected to boiling for 15 min. For the detection of neddylated substrates, nonreducing SDS–PAGE was performed, and the lysis buffer did not contain DTT. The cell lysates were subjected to separation by SDS–PAGE on 7.5–12.5% acrylamide gels at 90 V for 25 min or 180 V for 55 min. The separated proteins were transferred to a 0.45-µM nitrocellulose membrane using Trans-Blot Turbo (Bio-Rad), and subsequently blocked with TBST/5% milk for 1 h. The membranes were then incubated with primary antibodies at a dilution of 1:1,000, followed by secondary anti-rabbit HRP or anti-mouse HRP antibodies. The signal was analyzed using Bio-Rad ChemiDoc Imaging System.

## RT–qPCR

RT–qPCR was done to determine mRNA levels of the indicated genes as described previously (20, 28). The relative mRNA levels of

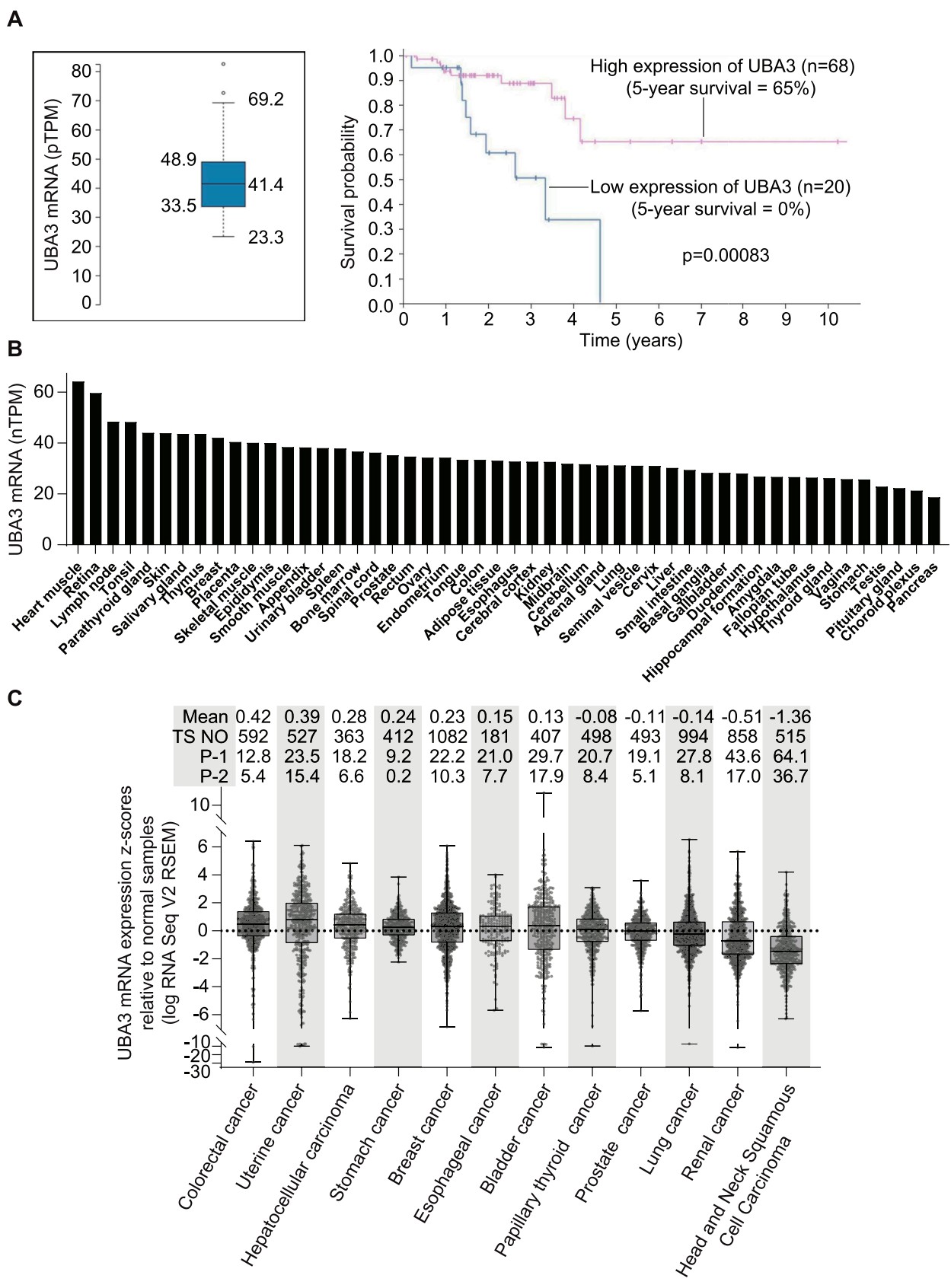

*NAE1* and *UBA3* were normalized to *GAPDH* transcripts. The primers used for *NAE1*, *UBA3*, and *GAPDH* are shown in Table S3 and synthesized by Sangon Biotech. All experiments were performed in triplicate and normalized to the *GAPDH* transcript levels using the comparative CT method.

### In vivo studies

In vivo studies were carried out as described previously ([2], [20]) following the guidelines of the Animal Care and Use Committee of the Shanghai Institute of *Materia Medica* and the institutional ethical guidelines. Xenograft models of human pancreatic cancer SUIT-2 cells or colon cancer RKO cells with or without *UBA3* deficiency or corresponding complementation were established in female BALB/c nude mice. Via oral administration, the mice were treated with various doses of SOMCL-19-133 or vehicle (5% DMAC + 5% Solutol HS15 + 90% ddH$_2$O with 0.5% MC) once daily (QD) for 21 consecutive days. The tumor volume was recorded twice weekly and was used to evaluate antitumor efficacy by calculating the relative tumor volume (RTV) and the treated/control ratio (T/C) on the final day.

### Statistical analyses

For the correlation analysis between UBA3 and NAE1 protein abundance in cancer cell lines, a scatter plot of UBA3 and NAE1 protein abundance was generated based on Cancer Cell Line Encyclopedia (CCLE; Broad, 2019), which was downloaded from cBioPortal.

For the statistical analyses of the mRNA levels of *UBA3* in cancer cell lines, data of *UBA3* mRNA were downloaded from the Human Protein Atlas (https://www.proteinatlas.org/). Noncancerous cell lines, uncategorized cell lines, and those cell lines with their number < 20 were excluded. The remaining 18 types of cancer cell lines (1,050) were listed from left to right according to their respective average levels of *UBA3* mRNA (mean).

For the retrospective analysis on the relationship between IC$_{50}$ values of SOMCL-19-133 in 28 cancer cell lines reported in reference [2] and corresponding cellular levels of *UBA3* mRNA from the Human Protein Atlas, the average *UBA3* mRNA levels and the average IC$_{50}$ values of SOMCL-19-133 in each group were calculated.

For the analysis of the mRNA levels of *UBA3* in tumor samples adapted from The Cancer Genome Atlas (TCGA) database, 6,961 human tumor samples with the corresponding data from their respective paracancerous normal tissues were included for further analysis. After excluding those types of tumor samples with their number < 100 and combining those samples from the same organs (but different tissues), the remaining 12 types of tumor samples were listed from left to right by their respective average levels of *UBA3* mRNA (mean).

All statistical data were assessed using GraphPad Prism 7.0 and presented as the mean ± SD unless otherwise stated. A *P*-value < 0.05 was considered to be statistically significant.*P* < 0.05; **P* < 0.01; ***P* < 0.001; and ns, not significant (*P* > 0.05).

## Data Availability

All data associated with this study are present in the article or the Supplementary Material.

## Supplementary Information

## Acknowledgements

This work was supported by a grant from the National Natural Science Foundation of China (82073875 to J-X He) and the State Key Laboratory of Drug Research.

### Author Contributions

Y-L Miao: conceptualization, data curation, formal analysis, validation, investigation, methodology, and writing—original draft, review, and editing.
L-N Zhou: conceptualization, data curation, formal analysis, validation, investigation, methodology, and writing—original draft.
S-S Song: data curation, formal analysis, validation, investigation, and methodology.
X-B Bao: data curation, formal analysis, validation, investigation, and methodology.
X-J Huan: data curation, formal analysis, validation, investigation, and methodology.
J Ding: conceptualization, resources, supervision, funding acquisition, and writing—review and editing.
J-X He: conceptualization, resources, supervision, funding acquisition, methodology, and writing—review and editing.

### Conflict of Interest Statement

The authors declare that they have no conflict of interest.

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
