## [Reviewer comments · Life Science Alliance]

UBA3 Reduction Sensitizes Cancer Cells to NAE Inhibitors

Yu-Ling Miao, Li-Na Zhou, Shan-Shan Song, Xubin Bao, Xiajuan Huan, Jian Ding, and Jinxue He

DOI: <https://doi.org/10.26508/lsa.202503589>

Corresponding author(s): Jinxue He, Shanghai Institute of Materia Medica

Review Timeline:	Submission Date:	2025-12-04
	Editorial Decision:	2026-02-06
	Revision Received:	2026-03-11
	Editorial Decision:	2026-04-05
	Revision Received:	2026-04-11
	Accepted:	2026-04-17

Scientific Editor: Sarita Hebbar

Transaction Report:

February 6, 2026

Re: Life Science Alliance manuscript #LSA-2025-03589-T

Yu-Ling Miao
Shanghai Institute of Materia Medica, Chinese Academy of Sciences
State Key Laboratory of Drug Research, Cancer Research Center
CHINA

Dear Dr. Miao,

Thank you for submitting your manuscript entitled "UBA3 Reduction Sensitizes Cancer Cells to NAE Inhibitors" to Life Science Alliance.

Your work was evaluated by two reviewers whose comments are appended below. As you will read the reviewers see a potential value in this work but have also raised several important points that preclude publication at this stage. Based on their overall evaluation, we invite you to submit a revised version.

We agree that you must address all of the concerns of Reviewer 2 related to the quantification of data (points 1, 8), and an accurate and more detailed description of results (points 4-7).

Although we concur that additional experiments to specifically rule out the involvement of autophagy in regulating NAE1/UBA3 protein stability (Rev 2 - point 2) and a limited proteomics dataset with the UB3 shRNA and NAE inhibitor treatment (Rev 1, suggested experiment) would strengthen this study, we leave it to your discretion to include this evidence. In case you do not provide these additional data, we agree that you must acknowledge these points whilst discussing the results.

A suitable revision should include additional changes to the text to resolve concerns made by Reviewer 1 points 1 and 2, and Reviewer 2 in point 3 and point 9.

I would be happy to discuss the revision in more detail via email or phone/videoconferencing. Please let me know which option you prefer, if any.

While you are revising your manuscript, please also attend to the below editorial points to help expedite the publication of your manuscript. Please direct any editorial questions to the journal office. When submitting the revision, please include a letter addressing the reviewers' comments point by point.

Thank you for this interesting contribution to Life Science Alliance. We hope that the comments below will prove constructive as your work progresses, and we are looking forward to receiving your revised manuscript.

Sincerely,

Sarita Hebbar, PhD
Scientific Editor
Life Science Alliance
<http://www.lsjournal.org>

-- High-resolution figure, supplementary figure and video files uploaded as individual files: See our detailed guidelines for

preparing your production-ready images, <https://www.life-science-alliance.org/authors>

B. MANUSCRIPT ORGANIZATION AND FORMATTING:

Reviewer #1 (Comments to the Authors (Required)):

Summary:

In this study, the authors investigate how the expression levels of the NAE subunits NAE1 and UBA3 influence cellular sensitivity to NAE inhibitors (NAEis) in cancer therapy. They first demonstrate that depletion of NAE1 reduces UBA3 levels and vice versa, indicating that the two subunits are interdependent. Using both in vitro and in vivo models, they show that UBA3 knockdown increases sensitivity to NAEis, an effect that is rescued by UBA3 re-expression, whereas exogenous UBA3 overexpression decreases cellular sensitivity to NAE inhibition. Furthermore, analysis of publicly available datasets and data from their previous work reveals an association between low UBA3 mRNA expression and increased sensitivity to NAEis across cancer cell lines. Consistently, xenograft models of the human pancreatic cancer cell line SUIT-2, which exhibits low UBA3 mRNA and protein levels, display enhanced sensitivity to NAE inhibition both in vitro and in vivo, although this relationship is not strictly proportional. Overall, the study proposes UBA3 expression as a candidate predictive biomarker for guiding NAE inhibitor-based cancer therapy. Although multiple NAE-targeting inhibitors have been developed, their clinical translation has been limited by dose-limiting toxicity and variable therapeutic efficacy. By demonstrating that reduced UBA3 expression increases cellular and in vivo sensitivity to NAE inhibition, the authors provide a potential stratification strategy to identify patients with low UBA3 levels more likely to benefit from lower, less toxic drug doses. This work therefore advances the field by offering a rational, biomarker-driven approach to improve the therapeutic index and precision of NAE-targeted anticancer therapies.

Main Comment

The manuscript is original, well written, and the data presented are generally strong and supportive of the authors' conclusions. The findings are of interest and represent a potential advance in the field. One aspect that would benefit from further strengthening-both in the text and, where feasible, with additional data-is a more detailed mechanistic explanation of how reduced UBA3 levels influence not only sensitivity to NAE inhibitors but also the downstream effects on NAE substrates and associated signaling pathways. Although Figure 6C presents a model for how UBA3 and NAE1 may regulate sensitivity to NAE inhibitors, expanding the discussion of the underlying molecular mechanisms would be valuable. In particular, the authors could clarify whether the increased sensitivity observed at low UBA3 levels arises solely from altered formation of the inhibitor-NAE adduct, or whether it also reflects broader changes in the regulation of NAE substrates and downstream cellular pathways. Even at the level of well-reasoned hypotheses or speculative models, such discussion would enhance the conceptual rigor of the study, clarify the biological basis of the observed phenotypes, and strengthen the translational relevance of the proposed biomarker.

Main points of the paper

1. Interdependence of NAE1 and UBA3 expression. The conclusion that NAE1 and UBA3 expression is interdependent is well supported by the data presented in this section. However, clarification is needed regarding the statement: "Treatment with the proteasome inhibitor MG132 increased the protein levels of NAE1 in control (shCtrl) cells but not in cells transfected with either shUBA3 or shNAE1 (Fig. 1D and E)." Based on Figure 1D, NAE1 levels also appear to increase following MG132 treatment in shUBA3 cells. The authors should reconcile this discrepancy between the text and the figure or revise the statement accordingly.
2. UBA3 reduction confers greater sensitization to NAE inhibitors than NAE1 reduction. This section represents a central pillar of the manuscript, and the data are clearly presented and largely supportive of the authors' claims. However, the reported increase

in downstream effector molecules is not sufficiently contextualized in the text. Given the central role of UBA3 and NAE1 in determining sensitivity to NAE inhibitors, a more explicit explanation of the biological significance of these effector changes would improve clarity. Specifically, the authors should discuss how these molecular alterations relate to pathway modulation and contribute to enhanced drug sensitivity.

3. Remaining main points. The final three main points are well supported by the data presented, and no major concerns are identified in these sections.

Additional Issues

In the Introduction, a concise description of the molecular function of the NEDD8-activating enzyme within the ubiquitin-proteasome pathway would improve accessibility for a broader readership. Specifically, outlining how NAE-mediated neddylation regulates cullin-RING ligase activity and downstream substrate degradation would provide important context for understanding why modulation of this pathway is particularly relevant in cancer biology.

Suggested Additional Experiments

While not essential for all claims, targeted proteomic analysis in selected experimental conditions (e.g., UBA3 knockdown combined with NAE inhibitor treatment) would substantially strengthen the manuscript. Such data could help identify altered pathways and molecular networks underlying the observed sensitization phenotype and provide mechanistic insight into how reduced UBA3 expression modulates cellular responses to NAE inhibition. Given that the authors propose UBA3 as a candidate biomarker for guiding therapeutic decisions, understanding which additional genes and pathways are perturbed under low UBA3 conditions would also enhance the translational relevance and safety considerations of this approach. This is particularly pertinent in light of the authors' conclusion that UBA3 expression and NAE inhibitor sensitivity are not strictly proportional, suggesting that multiple factors contribute to drug response. Even limited, focused proteomic profiling in a small number of representative models would therefore add meaningful depth and rigor to the study.

Reviewer #2 (Comments to the Authors (Required)):

Miao YL et al. present a comprehensive and mechanistically well-supported study demonstrating that reduced expression of the NAE catalytic subunit UBA3 markedly enhances cancer cell sensitivity to NEDD8-activating enzyme (NAE) inhibitors, both in vitro and in vivo. The authors further propose low UBA3 expression as a predictive biomarker for NAE inhibitor responsiveness, with potential implications for patient stratification and dose optimization. The study is technically sound and addresses an important question in translational cancer pharmacology- why NAE inhibitors show heterogeneous efficacy and dose-limiting toxicity in clinical settings. Overall, this is a strong and impactful study. However, the following points should be addressed.

- 1) Cul1 and its neddylated form are shown, but not quantitatively analyzed. Given the central role of NAE activity in Cullin neddylation, densitometric quantification of neddylated versus unneddylated Cul1 would substantially strengthen the functional interpretation.
- 2) The MG132 and BafA1 experiments provide useful preliminary insights but do not definitively exclude involvement of the proteasome or autophagy in regulating NAE1/UBA3 protein stability. Additional supporting evidence or, alternatively, a brief discussion acknowledging other possible mechanisms (e.g., complex stability or co-translational regulation) would improve this section.
- 3) The authors report that CRISPR/Cas9-generated UBA3 knockout cells retain approximately 30% residual UBA3 protein and attribute this to the essential nature of UBA3. As such, the term "UBA3 knockout" is potentially misleading, and these cells may be more accurately described as UBA3-deficient. Genomic validation should be provided, or the terminology should be revised accordingly. The implications of residual UBA3 expression for data interpretation should also be clarified.
- 4) In figure 3, rather than using separate graphs, a direct comparison graph showing UBA3 overexpression versus NAE1 overexpression would strengthen the conclusion that UBA3 is the dominant determinant of NAE inhibitor sensitivity.
- 5) The xenograft data in Figure 4 are compelling, particularly the apparent ~4-fold dose shift in UBA3-deficient tumors. While relative growth rate values are shown, full tumor growth curves over time are not always emphasized. Including complete tumor volume plots in supplementary figures would help readers assess growth kinetics and variability. In addition, pharmacokinetic equivalence across models is assumed but not demonstrated. Body weight alone may not fully capture systemic toxicity; if biochemical toxicity markers (e.g., liver enzymes) were not assessed, this should be explicitly stated. The authors should clarify whether drug exposure was comparable across groups and briefly discuss potential host-dependent effects.
- 6) The statistical tests used for IC₅₀ comparisons and xenograft analyses should be explicitly stated. Exact n values for animal experiments should be clearly provided, as mouse numbers per group are not always specified in the current figure legends.
- 7) In Figure 5 expression thresholds are emphasized rather than correlations. This could be misinterpreted as linear. Add a scatter plot of UBA3 mRNA vs IC₅₀ with regression and R² to improve clarity.
- 8) In Figure 5C, the relationship between UBA3 mRNA and protein levels is shown qualitatively. Quantitative correlation analysis across cell lines would strengthen this conclusion.
- 9) Although the study proposes UBA3 as a biomarker, there is limited discussion of how this would be implemented clinically, especially given prior failures of NAE inhibitors in late-stage trials. mRNA-based stratification may not reflect functional protein abundance. Tumor heterogeneity and normal tissue expression are not addressed. A more cautious and forward-looking discussion would help to contextualize clinical feasibility.

Minor comments:

1. Ensure consistent notation of SOMCL-19-133 vs. S133 throughout.
2. The manuscript references CCLE and Protein Atlas data interchangeably. Please clarify the source of protein abundance measurements (e.g., RPPA, mass spectrometry, or immunohistochemistry) and the normalization method used.
3. Some quantified panels lack clarity on n (biological vs technical replicates). Clearly state "n = how many independent biological replicates" in every relevant figure legend.

Point-by-point Responses

Editor's Comments:

We agree that you must address all of the concerns of Reviewer 2 related to the quantification of data (points 1, 8), and an accurate and more detailed description of results (points 4-7).

Response: We have done all as suggested.

Although we concur that additional experiments to specifically rule out the involvement of autophagy in regulating NAE1/UBA3 protein stability (Rev 2 - point 2) and a limited proteomics dataset with the UB3 shRNA and NAE inhibitor treatment (Rev 1, suggested experiment) would strengthen this study, we leave it to your discretion to include this evidence. In case you do not provide these additional data, we agree that you must acknowledge these points whilst discussing the results.

Response: We have made additional detailed explanation and discussion on these issues.

A suitable revision should include additional changes to the text to resolve concerns made by Reviewer 1 points 1 and 2, and Reviewer 2 in point 3 and point 9.

Response: We have done all as suggested.

Reviewer #1 (Comments to the Authors (Required)):

Summary:

In this study, the authors investigate how the expression levels of the NAE subunits NAE1 and UBA3 influence cellular sensitivity to NAE inhibitors (NAEIs) in cancer therapy. They first demonstrate that depletion of NAE1 reduces UBA3 levels and vice versa, indicating that the two subunits are interdependent. Using both in vitro and in vivo models, they show that UBA3 knockdown increases sensitivity to NAEIs, an effect that is rescued by UBA3 re-expression, whereas exogenous UBA3 overexpression decreases cellular sensitivity to NAE inhibition. Furthermore, analysis of publicly

available datasets and data from their previous work reveals an association between low UBA3 mRNA expression and increased sensitivity to NAEis across cancer cell lines. Consistently, xenograft models of the human pancreatic cancer cell line SUIT-2, which exhibits low UBA3 mRNA and protein levels, display enhanced sensitivity to NAE inhibition both in vitro and in vivo, although this relationship is not strictly proportional. Overall, the study proposes UBA3 expression as a candidate predictive biomarker for guiding NAE inhibitor-based cancer therapy. Although multiple NAE-targeting inhibitors have been developed, their clinical translation has been limited by dose-limiting toxicity and variable therapeutic efficacy. By demonstrating that reduced UBA3 expression increases cellular and in vivo sensitivity to NAE inhibition, the authors provide a potential stratification strategy to identify patients with low UBA3 levels more likely to benefit from lower, less toxic drug doses. This work therefore advances the field by offering a rational, biomarker-driven approach to improve the therapeutic index and precision of NAE-targeted anticancer therapies.

Response: Thank the reviewer for the positive comments.

Main Comment

The manuscript is original, well written, and the data presented are generally strong and supportive of the authors' conclusions. The findings are of interest and represent a potential advance in the field. One aspect that would benefit from further strengthening-both in the text and, where feasible, with additional data-is a more detailed mechanistic explanation of how reduced UBA3 levels influence not only sensitivity to NAE inhibitors but also the downstream effects on NAE substrates and associated signaling pathways. Although Figure 6C presents a model for how UBA3 and NAE1 may regulate sensitivity to NAE inhibitors, expanding the discussion of the underlying molecular mechanisms would be valuable. In particular, the authors could clarify whether the increased sensitivity observed at low UBA3 levels arises solely from altered formation of the inhibitor-NAE adduct, or whether it also reflects broader changes in the regulation of NAE substrates and downstream cellular pathways. Even at the level of well-reasoned hypotheses or speculative models, such discussion would

enhance the conceptual rigor of the study, clarify the biological basis of the observed phenotypes, and strengthen the translational relevance of the proposed biomarker.

Response: Thank the reviewer for the positive comments and excellent suggestions. We have added additional discussions about how reduced UBA3 levels influence the downstream effects on NAE substrates and associated signaling pathways in the section of Discussion.

Main points of the paper

1. Interdependence of NAE1 and UBA3 expression. The conclusion that NAE1 and UBA3 expression is interdependent is well supported by the data presented in this section. However, clarification is needed regarding the statement: "Treatment with the proteasome inhibitor MG132 increased the protein levels of NAE1 in control (shCtrl) cells but not in cells transfected with either shUBA3 or shNAE1 (Fig. 1D and E)." Based on Figure 1D, NAE1 levels also appear to increase following MG132 treatment in shUBA3 cells. The authors should reconcile this discrepancy between the text and the figure or revise the statement accordingly.

Response: We apologize for any confusions caused. We have carefully checked the text, figures and data, and revised the statement accordingly.

2. UBA3 reduction confers greater sensitization to NAE inhibitors than NAE1 reduction. This section represents a central pillar of the manuscript, and the data are clearly presented and largely supportive of the authors' claims. However, the reported increase in downstream effector molecules is not sufficiently contextualized in the text. Given the central role of UBA3 and NAE1 in determining sensitivity to NAE inhibitors, a more explicit explanation of the biological significance of these effector changes would improve clarity. Specifically, the authors should discuss how these molecular alterations relate to pathway modulation and contribute to enhanced drug sensitivity.

Response: Thank the reviewer for the excellent suggestion. We have added additional background descriptions in the section of Introduction and more discussions in the section of Discussion.

3. Remaining main points. The final three main points are well supported by the data

presented, and no major concerns are identified in these sections.

Response: Thanks.

Additional Issues

In the Introduction, a concise description of the molecular function of the NEDD8-activating enzyme within the ubiquitin-proteasome pathway would improve accessibility for a broader readership. Specifically, outlining how NAE-mediated neddylation regulates cullin-RING ligase activity and downstream substrate degradation would provide important context for understanding why modulation of this pathway is particularly relevant in cancer biology.

Response: Thank the reviewer for the excellent suggestion. We have added additional background descriptions as suggested in the section of Introduction

Suggested Additional Experiments

While not essential for all claims, targeted proteomic analysis in selected experimental conditions (e.g., UBA3 knockdown combined with NAE inhibitor treatment) would substantially strengthen the manuscript. Such data could help identify altered pathways and molecular networks underlying the observed sensitization phenotype and provide mechanistic insight into how reduced UBA3 expression modulates cellular responses to NAE inhibition. Given that the authors propose UBA3 as a candidate biomarker for guiding therapeutic decisions, understanding which additional genes and pathways are perturbed under low UBA3 conditions would also enhance the translational relevance and safety considerations of this approach. This is particularly pertinent in light of the authors' conclusion that UBA3 expression and NAE inhibitor sensitivity are not strictly proportional, suggesting that multiple factors contribute to drug response. Even limited, focused proteomic profiling in a small number of representative models would therefore add meaningful depth and rigor to the study.

Response: We agree that “targeted proteomic analysis in selected experimental conditions (e.g., UBA3 knockdown combined with NAE inhibitor treatment) would substantially strengthen the manuscript”. So we searched for the literature about proteomic analyses in different conditions with NAE inhibitor treatments and found

five related reports (i.e., Mol Cell Proteomics, 2011,10:M111.009183; Mol Cell Proteomics, 2013,12:2370-80; Proteomics, 2014,14:2008-16; Leukemia, 2016,30:1190-4; and J Proteome Res, 2019;18:1893-1901). The work mentioned above used different cells, including A375 (melanoma; 2011), HUVEC (human umbilical vein endothelial cells; 2013 and 2014), MV4-11 (leukemia; 2016), and HeLa (cervical cancer; 2019), all of which were treated with the NAE inhibitor MLN4924. The above work also used different methods to conduct proteomic analyses, including stable isotope labeling with amino acids (SILAC) plus LC/MS/MS (2011), a Fast-seq workflow of the use of dual reverse phase HPLC-MS (2013), a rapid and reproducible 1D LC-MS/MS workflow (2014), shotgun proteomics-comprehensive protein profiling (2016), and SWARM (2019). Comparison of the results from those reports showed that identical proteins quantitatively changed by MLN4924 treatments were primarily involved in regulation of cell death and survival, gene expression, cell cycle, and DNA replication, recombination and repair. Moreover, several critical issues were also shown as following. Firstly, quantitative changes in proteins (types and degrees) are highly dependent of the MLN4924 exposure duration; secondly, those changes are dependent of tumor or cell types (for example, only 36% of the proteins that were pharmacodynamically increased by MLN4924 were identical between two separate analyses in MV4-11 AML cells and in A375 melanoma cells); and finally, the results from different methods to conduct proteomic analyses varied largely, even in the same cells, i.e., HUVEC (2013 versus 2014). These issues, particularly in addition to UBA3 knockdown, are likely to make it more complicated to identify common altered pathways and molecular networks underlying the observed sensitization phenotype and to provide common mechanistic insights into how reduced UBA3 expression modulates cellular responses to NAE inhibition. Therefore, additional independent project(s) might be more suitable to answer this comment from the reviewer.

Reviewer #2 (Comments to the Authors (Required)):

Miao YL et al. present a comprehensive and mechanistically well-supported study demonstrating that reduced expression of the NAE catalytic subunit UBA3 markedly

enhances cancer cell sensitivity to NEDD8-activating enzyme (NAE) inhibitors, both in vitro and in vivo. The authors further propose low UBA3 expression as a predictive biomarker for NAE inhibitor responsiveness, with potential implications for patient stratification and dose optimization. The study is technically sound and addresses an important question in translational cancer pharmacology- why NAE inhibitors show heterogeneous efficacy and dose-limiting toxicity in clinical settings. Overall, this is a strong and impactful study. However, the following points should be addressed.

Response: Thank the reviewer for the positive comments.

1) Cull1 and its neddylylated form are shown, but not quantitatively analyzed. Given the central role of NAE activity in Cullin neddylation, densitometric quantification of neddylylated versus unneddylylated Cull1 would substantially strengthen the functional interpretation.

Response: Thank the reviewer for the insightful suggestion. We have done densitometric quantification of neddylylated versus unneddylylated Cull1, and the data were presented in Fig. S1. Correspondingly, we have also revised the descriptions in the section of Results.

2) The MG132 and BafA1 experiments provide useful preliminary insights but do not definitively exclude involvement of the proteasome or autophagy in regulating NAE1/UBA3 protein stability. Additional supporting evidence or, alternatively, a brief discussion acknowledging other possible mechanisms (e.g., complex stability or co-translational regulation) would improve this section.

Response: We have integrated the reviewer's insightful suggestion into our revised manuscript (in the section of Results).

3) The authors report that CRISPR/Cas9-generated UBA3 knockout cells retain approximately 30% residual UBA3 protein and attribute this to the essential nature of UBA3. As such, the term "UBA3 knockout" is potentially misleading, and these cells may be more accurately described as UBA3-deficient. Genomic validation should be provided, or the terminology should be revised accordingly. The implications of residual UBA3 expression for data interpretation should also be clarified.

Response: As suggested, we have replaced "UBA3 knockout" with UBA3-deficient or

UBA3 deficiency, where applicable. We also made further clarification about the implications of residual UBA3 expression in the section of Discussion.

4) In figure 3, rather than using separate graphs, a direct comparison graph showing UBA3 overexpression versus NAE1 overexpression would strengthen the conclusion that UBA3 is the dominant determinant of NAE inhibitor sensitivity.

Response: As suggested, we have used a direct comparison graph to compare the effects of UBA3 overexpression and NAE1 overexpression on NAE inhibitor sensitivity (revised Figure 3B).

5) The xenograft data in Figure 4 are compelling, particularly the apparent ~4-fold dose shift in UBA3-deficient tumors. While relative growth rate values are shown, full tumor growth curves over time are not always emphasized. Including complete tumor volume plots in supplementary figures would help readers assess growth kinetics and variability. In addition, pharmacokinetic equivalence across models is assumed but not demonstrated. Body weight alone may not fully capture systemic toxicity; if biochemical toxicity markers (e.g., liver enzymes) were not assessed, this should be explicitly stated. The authors should clarify whether drug exposure was comparable across groups and briefly discuss potential host-dependent effects.

Response: Thank the reviewer for the excellent suggestion and careful consideration. We have added full tumor growth curves over time (Supplementary Fig. 2) and made corresponding revisions accordingly. We also added an additional paragraph in the section of Results as follows.

“In the above experiments, comparable nude mouse models were used. Nevertheless, pharmacokinetic equivalence across models and comparable drug exposure across groups were not demonstrated. Potential host-dependent effects were not assessed. Additionally, biochemical toxicity markers (e.g., liver enzymes) were not measured, and changes in body weight of mice alone may not fully capture systemic toxicity. Therefore, caution is warranted when interpreting these results.”

6) The statistical tests used for IC₅₀ comparisons and xenograft analyses should be explicitly stated. Exact n values for animal experiments should be clearly provided, as mouse numbers per group are not always specified in the current figure legends.

Response: Thank the reviewer for the suggestion. We have stated the statistical tests in the analyses and provided the n values for animal experiments in the revised figure legends.

7) In Figure 5 expression thresholds are emphasized rather than correlations. This could be misinterpreted as linear. Add a scatter plot of UBA3 mRNA vs IC₅₀ with regression and R² to improve clarity.

Response: We appreciate the reviewer's thoughtful suggestion. We have added a scatter plot of UBA3 mRNA vs IC₅₀ with regression and R² related to Fig.5B (Supplementary Fig. 3).

8) In Figure 5C, the relationship between UBA3 mRNA and protein levels is shown qualitatively. Quantitative correlation analysis across cell lines would strengthen this conclusion.

Response: A quantitative correlation analysis between UBA3 mRNA and protein levels across cell lines was carried out and the figure legends were revised correspondingly.

9) Although the study proposes UBA3 as a biomarker, there is limited discussion of how this would be implemented clinically, especially given prior failures of NAE inhibitors in late-stage trials. mRNA-based stratification may not reflect functional protein abundance. Tumor heterogeneity and normal tissue expression are not addressed. A more cautious and forward-looking discussion would help to contextualize clinical feasibility.

Response: Thank the reviewer for the careful consideration. We have added additional discussion in the section of Discussion accordingly.

Minor comments:

1. Ensure consistent notation of SOMCL-19-133 vs. S133 throughout.

Response: We have made corresponding revisions.

2. The manuscript references CCLE and Protein Atlas data interchangeably. Please clarify the source of protein abundance measurements (e.g., RPPA, mass spectrometry, or immunohistochemistry) and the normalization method used.

Response: We are so sorry for this mistake. In Fig.1A, we used the protein abundance ratios relative to bridge-sample data of UBA3 and NAE1 from CCLE (Broad, 2019) for correlation analysis. The source of protein abundance measurements is mass spectrometry. In Fig.5 and Fig.7, we used the Human Protein Atlas (<https://www.proteinatlas.org/>) to obtain *UBA3* mRNA data in cancer cell lines, patients with rectum adenocarcinoma, and human normal tissues. We have made corresponding revisions throughout the manuscript. The Human Protein Atlas used RNA-seq to quantify mRNA levels.

3. Some quantified panels lack clarity on n (biological vs technical replicates). Clearly state "n = how many independent biological replicates" in every relevant figure legend.

Response: Thank the reviewer for the suggestion. We have revised the relevant figure legends accordingly.

April 5, 2026

RE: Life Science Alliance Manuscript #LSA-2025-03589-TR

Dr. Yu-Ling Miao
Shanghai Institute of Materia Medica
State Key Laboratory of Drug Research, Cancer Research Center
501 Haik Road
Shanghai 201203
China

Dear Dr. Miao,

Thank you for submitting your revised manuscript entitled "UBA3 Reduction Sensitizes Cancer Cells to NAE Inhibitors". Your manuscript was evaluated by both the original reviewers whose notes are appended below.

As you will read, both reviewers find the revised version has addressed their concerns and is improved.

Overall, In line with the reviewers' evaluation, we would be happy to publish your paper in Life Science Alliance pending final revisions necessary to meet our formatting guidelines.

MANUSCRIPT ORGANIZATION AND FORMATTING:

To avoid unnecessary delays in the acceptance and publication of your paper, please read the following information carefully. Full guidelines are available on our Instructions for Authors page, <https://www.life-science-alliance.org/authors>

- The abstract provided in the manuscript document is over the allowed limit 175 words. Please follow our guidelines and modify your abstract (<https://www.life-science-alliance.org/manuscript-prep#format>).
- Please include a scale bar for images in Figure 4.
- Please reorganise information in the legends for Figures 5 and 7 such the methodological details are included in a sub-section in materials and methods. Further please follow our guidelines to modify this legend (<https://www.life-science-alliance.org/manuscript-prep#legends>).
- Please cite a source for the structures shown in Figure 6A in the legends.
- Please modify the legend for Supplementary Figure 1, to modify the statement "In this figure, (A) and (B) were quantitatively equivalent to Fig. 1(B) and Fig. 2(C), respectively." to "In this figure, (A) and (B) are quantifications for the blots in Fig. 1(B) and Fig. 2(C) respectively.
- Thank you for providing a citation for SOMCL-19-133. Please expand on the preparation for this work.
- Please also expand the methods section for the description of (1) Neddylation of Cul1 in Figure 1B, (2) correlation presented in Figure 1A including the protein source (3) quantification applied for qRT-PCR experiments, (4) antibodies used in Western blotting for verification of reduced protein following shRNA or CRISPR-Cas9 gene targeting, (5) lenti-viral based infection for exogenous expression of UBA3 and NAE1, and (6) use of sulforhodamine B (SRB) in cell viability and CCK-8 assays. Kindly include a source for SRB.
- Please remove the separate supporting information file. Tables should be uploaded in an editable format, separately. Supplementary legends should be provided in the manuscript file after the main figure legends.
- Please add a Running Title and the abstract in our system. Abstract in the system and the manuscript file must match.
- Please add ORCID ID for the corresponding author - they should have received instructions on how to do so.
- Please add a Summary Blurb/Alternate Abstract in our system.
- Please add Keywords and category for your manuscript in our system.
- Please add the X and Bluesky handles of your host institute/organisation, as well as your own, and/or one of the authors, in our system.
- Corresponding Author must match between the system and the manuscript file.
- Please upload a clean manuscript file without the coloured text. The version with highlighted changes, you should upload as a related manuscript file.
- The contributions selected for Jian Ding do not qualify them for authorship. Please either update the contributions in our system and in the Author Contributions section of the manuscript, or let us know if the authors need to be removed (and added eventually to the acknowledgment section).
- Please be sure that the authorship listing and order is correct.

We welcome submissions of potential cover images for the issue of LSA in which your work would appear. If you have high quality images associated with this work, please feel free to email these, with a caption, to the journal office.

LSA encourages authors to provide a 30-60 second video where the study is briefly explained. We will use these videos on social media to promote the published paper and the presenting author (for examples, see <https://docs.google.com/document/d/1-UWCfbE4pGcDdcgzcmiuJI2XMBJnxKYeqRvLLrLSo8s/edit?usp=sharing>). Corresponding or first-authors are welcome to submit the video. Please submit only one video per manuscript. The video can be emailed to contact@life-science-alliance.org

FINAL FILES:

The following items are required for acceptance.

The license to publish form must be signed before your manuscript can be sent to production. A link to the license to publish form will be available to the corresponding author only. Please take a moment to check your funder requirements.

Thank you for your attention to these final processing requirements. Please revise and format the manuscript and upload materials as soon as you are able.

Thank you for this interesting contribution to the literature. We look forward to publishing your paper in Life Science Alliance.

Sincerely,

Sarita Hebbar, PhD
Scientific Editor
Life Science Alliance
<http://www.lsjournal.org>

Reviewer #1 (Comments to the Authors (Required)):

I have reviewed the revised manuscript and appreciate the authors' thoughtful and thorough responses to the comments provided in the previous round of revisions.

I acknowledge that overall the comments were addressed effectively. The revisions substantially improve the manuscript, and I

am pleased to recommend it for publication without further comment.

Reviewer #2 (Comments to the Authors (Required)):

Miao YL et al. have addressed all of my major and minor comments thoroughly. They have included the quantitative analysis I asked for, added appropriate statistical details and sample sizes, improved the data presentation including new comparison graphs and supplementary tumor growth curves, and incorporated additional correlation analysis. The clarification of experimental limitations and expansion of the discussion regarding clinical translation and alternative mechanisms, have significantly strengthened the manuscript. Where additional experiments were not performed, the authors have provided reasonable justification and have acknowledged the limitations in the discussion. Overall, the revised manuscript is substantially improved. I therefore recommend acceptance of the manuscript in its current form.

April 17, 2026

RE: Life Science Alliance Manuscript #LSA-2025-03589-TRR

Jinxue He
Shanghai Institute of Materia Medica
CHINA

Dear Dr. He,

Thank you for submitting your Research Article entitled "UBA3 Reduction Sensitizes Cancer Cells to NAE Inhibitors". It is a pleasure to let you know that your manuscript is now accepted for publication in Life Science Alliance. Congratulations on this interesting work.

Your article will publish open access upon publication under a CC-BY license.

DISTRIBUTION OF MATERIALS:

Again, congratulations on a very nice paper. I hope you found the review process to be constructive and are pleased with how the manuscript was handled editorially. We look forward to future exciting submissions from your lab.

Sincerely,

Sarita Hebbar, PhD
Scientific Editor
Life Science Alliance
<http://www.lsajournal.org>